# Ultrathin-metal-film-based transparent electrodes with relative transmittance surpassing 100%

Chengang Ji [1,5], Dong Liu [2,5✉], Cheng Zhang[3,4] & L. Jay Guo [1✉]

Flexible transparent electrodes are in significant demand in applications including solar cells, light-emitting diodes, and touch panels. The combination of high optical transparency and high electrical conductivity, however, sets a stringent requirement on electrodes based on metallic materials. To obtain practical sheet resistances, the visible transmittance of the electrodes in previous studies is typically lower than the transparent substrates the electrode structures are built on, namely, the transmittance relative to the substrate is <100%. Here, we demonstrate a flexible dielectric-metal-dielectric-based electrode with ~88.4% absolute transmittance, even higher than the ~88.1% transmittance of the polymer substrate, which results in a relative transmittance of ~100.3%. This non-trivial performance is achieved by leveraging an optimized dielectric-metal-dielectric structure guided by analytical and quantitative principles described in this work, and is attributed to an ultra-thin and ultra-smooth copper-doped silver film with low optical loss and low sheet resistance.

[1] Department of Electrical Engineering and Computer Science, University of Michigan, 1301 Beal Avenue, Ann Arbor, MI 48109, USA. [2] MIIT Key Laboratory of Thermal Control of Electronic Equipment, School of Energy and Power Engineering, Nanjing University of Science and Technology, 200 Xiaolingwei Street, 210094 Nanjing, China. [3] School of Optical and Electronic Information, Huazhong University of Science and Technology, 1037 Luoyu Road, 430074 Wuhan, China. [4] Wuhan National Laboratory for Optoelectronics, Huazhong University of Science and Technology, 1037 Luoyu Road, 430074 Wuhan, China. [5] These authors contributed equally: Chengang Ji, Dong Liu. ✉email: liudong15@njust.edu.cn; guo@umich.edu

Transparent electrodes are widely used in photovoltaics (PVs)[1,2], light-emitting diodes (LEDs)[3–5], touch panels[6,7], and other optoelectronic devices. Indium tin oxide (ITO) is the conventional selection for the transparent electrode because of its high visible transmittance and electrical conductivity. However, the low abundance of the indium element on earth is a limiting factor of this material. In addition, its applications in emerging flexible optoelectronic devices are significantly hindered by both the poor mechanical flexibility and the high annealing temperature needed to reduce its resistivity[8]. Graphene[9,10], carbon nanotubes (CNTs)[11,12], and conductive polymers[13] have been investigated to replace ITO. Unfortunately, their limited conductivity and high cost of mass-production remains a challenge in large-area applications. To overcome these limitations, transparent electrodes employing metal networks have been proposed[14–23]. Although these novel structures exhibit decent optical and electrical performance and can also be scaled up easily via different roll-to-roll (R2R) techniques[24,25], the extruded or non-uniform surfaces may cause shorting problems when being used in LED and PV devices[14]. In addition, the high optical haze resulting from the scattering of the nanowires and mesh patterns is undesired in high-resolution displays[14]. In recent years, dielectric–metal–dielectric (DMD)-based transparent electrodes have been noted as potential alternatives. In this type of electrode, a thin metallic film is sandwiched between two antireflection dielectrics to induce high transparency. They feature high transparency and conductivity, low haze, excellent flexibility, facile fabrication, and great compatibility with different substrates[2,26–37]. The trade-off between electrical conductivity and optical transmittance has been a major challenge for metallic-material-based transparent electrodes. To achieve a practical conductivity (e.g., sheet resistance ($R_s$) < 20 Ω sq$^{-1}$), the absolute visible transmittance of previously reported metallic-material-based transparent electrodes, including both metal network and DMD structures, is typically lower than that of the substrate itself, which has been taken for granted without serious questioning.

Therefore, we are motivated to conduct a rigorous investigation and explore the limit of transmittance at a sufficiently low sheet resistance suitable for practical applications. In this work, we demonstrate a DMD-based transparent electrode with ~88.4% absolute transmittance averaged over the entire visible spectrum (400–700 nm) on polyethylene terephthalate (PET) polymeric substrate, which surpasses the transmittance of the substrate itself (~88.1%), leading to a relative transmittance >100%. This counter-intuitive, yet achievable performance is obtained by (i) quantitative design principles that are generalized in this work, particularly, analytic expression of the optimal bottom dielectric thickness and analytical result showing that different materials should be used for the two dielectrics, (ii) the use of an ultra-thin (~6.5 nm thick) and ultra-smooth (roughness < 1 nm) copper-doped silver (Cu-doped Ag) film providing low optical loss and high electrical conductivity at the same time. The proposed design principles and electrode structures have resolved the problems faced by other existing transparent electrodes and may have the potential to replace traditional ITO counterparts for flexible optoelectronics, thus facilitating high-performance flexible displays and optoelectronic devices.

## Results

**Strategy to achieve relative transmittance surpassing 100%.**
Relative transmittance, $T_r$, is defined as the ratio of the absolute transmittance of the structure where a transparent substrate is coated with a metallic electrode, $T_D$ (see Fig. 1a), to the absolute transmittance of the bare substrate, $T_S$ (Fig. 1b):

$$T_r = \frac{T_D}{T_S} \approx \frac{1 - (R_1 + A) - R_2}{1 - R_2 - R_2}. \quad (1)$$

where $R_1$ is the reflection at the top side of DMD and $A$ is the absorption of the metallic film, and $R_2$ is the reflection at the substrate/air interface. For simplicity, the multiple round-trip reflections between the front and bottom surfaces of the substrate are ignored due to the negligible reflection intensity at both interfaces. Intuitively, $T_r$ is thought to be smaller than 100% because a metallic electrode is usually reflective. However, in this section, we will provide guidelines built around the antireflection principles to reduce $R_1$ and $A$ using the DMD structure, and show that with optimized design ($R_1 + A$) can be made smaller than the single side reflection of the substrate, $R_2$, leading to relative transmittance surpassing 100%.

As shown in Fig. 2a, seven parameters (the refractive index, $n_2$, and thickness, $d_2$, of the transparent Dielectric 1, the refractive index, $n_3$, extinction coefficient, $\kappa_3$, and thickness, $d_3$, of the metallic film, and the refractive index, $n_4$, and thickness, $d_4$, of the transparent Dielectric 2) should all be considered in order to optimize the transmittance over a broad spectrum in the visible, which makes it challenging to provide quantitative design guidelines[33–35]. Our design strategy is to maximize the transmittance for the condition where the metallic film is thick enough to achieve a practical sheet resistance; and thus, this strategy enables the synergistic combination of high optical transmittance and high electrical conductivity of DMD-based transparent electrodes.

As a first step in the design process, a suitable material is selected for the sandwiched metallic layer by considering its (i) electrical conductivity; (ii) light absorption property in the visible range. Among all metals, silver (Ag), copper (Cu), and

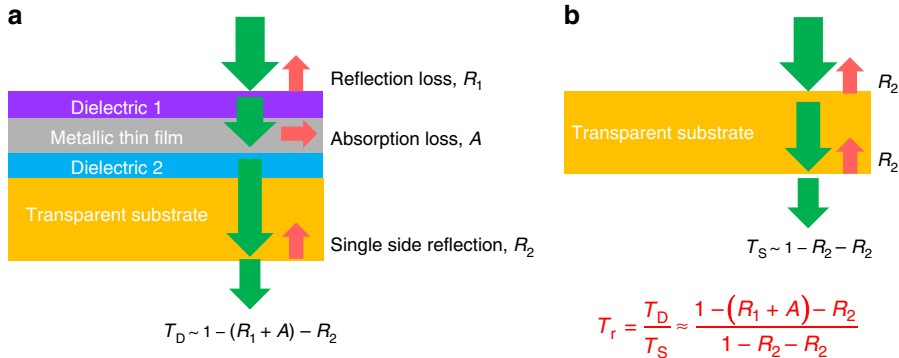

**Fig. 1 The definition of relative transmission. a** Transparent substrate coated with a DMD structure. **b** Bare substrate.

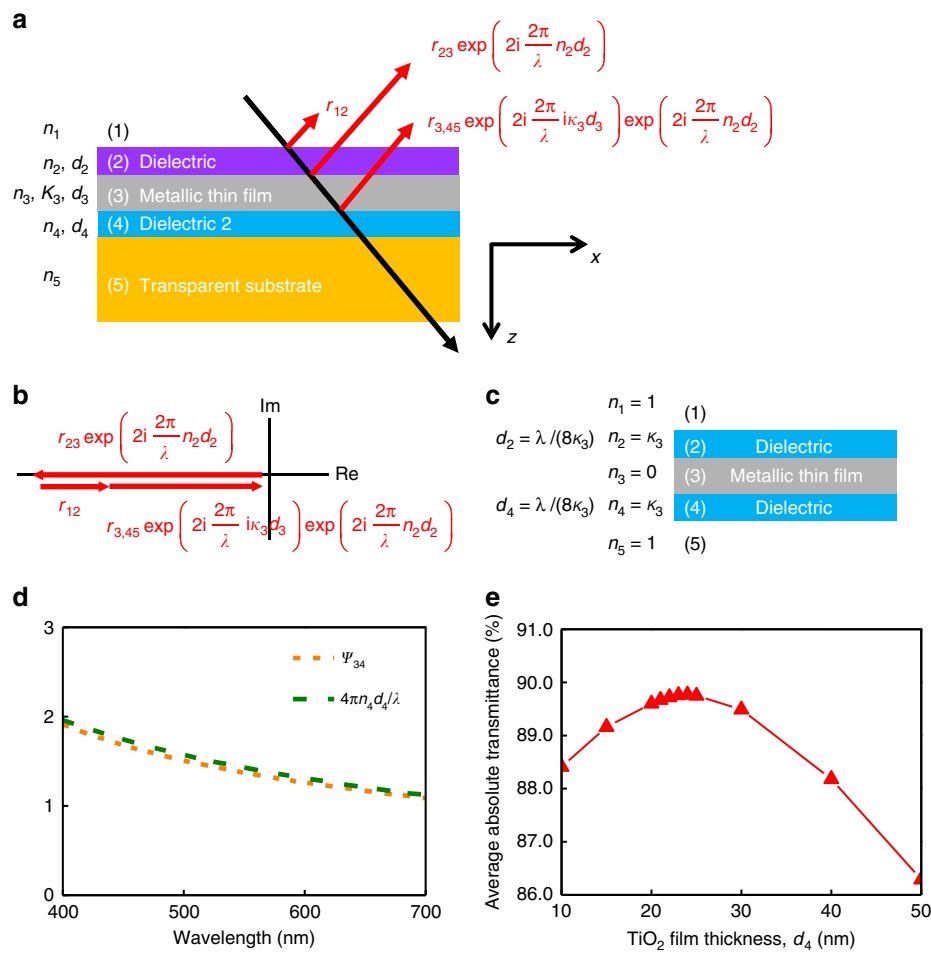

**Fig. 2 Theoretical design of a DMD electrode. a** Design parameters and reflection waves at various interfaces of a DMD transparent electrode. **b** Phasor diagram of reflection waves. **c** An ideal DMD and its parameters. **d** The calculated values of the left and right terms in Eq. (9) when using 24 nm TiO$_2$ and 9 nm Ag as Dielectric 2 and the metallic layer, respectively, in the DMD structure, showing that Eq. (9) is well satisfied across the entire visible range. **e** Optimal averaged transmittance of the DMD electrode across the visible range (400–700 nm) dependent on the thickness of Dielectric 2. Here, TiO$_2$ and 9 nm Ag are used as the bottom dielectric layer and the middle metallic layer, respectively. Each optimal transmittance value at a fixed TiO$_2$ thickness is found out by sweeping the refractive index ($n_2$) and the thickness ($d_2$) of Dielectric 1.

gold (Au) exhibit the lowest intrinsic electrical resistivity. The absorption of light by the metallic film in a DMD structure is expressed as

$$A(\lambda_0) = \frac{4\pi n_3 \kappa_3}{\lambda_0} \frac{\int_0^{d_3} |E(z)|^2 \mathrm{d}z}{|E_0|^2}, \qquad (2)$$

where $E(z)$ is the electric field in the metallic film, $E_0$ is the incident electric field and $\lambda_0$ is the free-space wavelength. With a given thickness of the metallic layer, a metal with a small ($nk$) should be chosen for low absorption and high transmittance. Among Ag, Cu, and Au, Ag ($n = 0.13$ and $k = 3.17$ at 550 nm wavelength[38]) offers the lowest ($nk$) in the visible range, and is therefore, employed as the metallic layer in this study. Considering the small real part refractive index of Ag, $n_3 = 0$ is reasonably assumed for the rest of this section for simplicity.

In the next step, the thickness and the refractive index of Dielectric 2 are determined. To reduce the reflection loss at the top surface, the sum of two vectors shown in Fig. 2a should be designed to cancel $r_{12}$, which is the complex reflection coefficient at the interface (1–2 interface) between the air ($n_1 = 1$) and Dielectric 1 ($r_{pq} = (\tilde{n}_p \cos\theta_p - \tilde{n}_q \cos\theta_q)/(\tilde{n}_p \cos\theta_p + \tilde{n}_q \cos\theta_q)$ is

the reflection coefficient at $p$–$q$ interface for TE polarization where $\tilde{n}_{p(q)} = n_{p(q)} + i\kappa_{p(q)}$ is the complex refractive index of layer $p(q)$ and $\theta_{p(q)}$ is the direction of wave propagation in the corresponding layer). The first vector is $r_{23}$ with a phase shift acquired from the wave propagation in Dielectric 1. The second is $r_{3,45}$ with a magnitude attenuation resulting from the propagation in the metallic film and with a phase shift due to the propagation in Dielectric 1. $r_{3,45}$ is the total reflection coefficient at 3–4 interface (where the light reflected from the interface between 4 and 5 is taken into account), and can be expressed as

$$r_{3,45} = \frac{r_{34} + r_{45} \exp\left(2i\frac{2\pi}{\lambda}n_4 d_4\right)}{1 + r_{34} r_{45} \exp\left(2i\frac{2\pi}{\lambda}n_4 d_4\right)}. \qquad (3)$$

In order to cancel out each other yielding $R_1 = 0$, the trajectory of these three vectors should be a triangle or they should be aligned in the way as shown in Fig. 2b, therefore

$$\left| r_{3,45} \exp\left(2i\frac{2\pi}{\lambda}i\kappa_3 d_3\right) \exp\left(2i\frac{2\pi}{\lambda}n_2 d_2\right) \right| \geq \left| r_{23} \exp\left(2i\frac{2\pi}{\lambda}n_2 d_2\right) \right| - |r_{12}|. \qquad (4)$$

Since $n_3 = 0$ as we assumed,

$$\left| r_{23} \exp\left( 2i \frac{2\pi}{\lambda} n_2 d_2 \right) \right| = 1 > |r_{12}|, \tag{5}$$

$$\left| r_{23} \exp\left( 2i \frac{2\pi}{\lambda} n_2 d_2 \right) \right| = 1 > \left| r_{3,45} \exp\left( 2i \frac{2\pi}{\lambda} i\kappa_3 d_3 \right) \exp\left( 2i \frac{2\pi}{\lambda} n_2 d_2 \right) \right|, \tag{6}$$

Thus, the right term in Eq. (4) represents the minimum value of the left term, which is achieved when the three vectors are aligned in the way as shown in Fig. 2b. In this case, we have

$$\left| r_{3,45} \exp\left( 2i \frac{2\pi}{\lambda} i\kappa_3 d_3 \right) \exp\left( 2i \frac{2\pi}{\lambda} n_2 d_2 \right) \right| = \left| r_{23} \exp\left( 2i \frac{2\pi}{\lambda} n_2 d_2 \right) \right| - |r_{12}|, \tag{7}$$

When the left term in Eq. (4) is smaller than this minimum value, the reflection cannot be completely suppressed.

Since the left term in Eq. (4) decreases with increasing metallic film thickness ($d_3$), $|r_{3,45}|$ should be designed to achieve its maximum value so that Eq. (4) can be satisfied for maximum metallic film thickness to achieve high electrical conductivity. To maximize $|r_{3,45}|$, $r_{34}$ and $r_{45}$ should interfere constructively with each other so that

$$\psi_{prop} + \psi_{45} - \psi_{34} = 0, \tag{8}$$

where $\psi_{34}$ is the phase angle of $r_{34}$, representing the phase shift due to the reflection at 3–4 interface, and $\psi_{45}$ is the phase angle of $r_{45}$, representing the reflection phase shift at 4–5 interface. $n_4 > n_5$ (shown later) gives $\psi_{45} = 0$. $\psi_{prop} = 4\pi n_4 d_4/\lambda$ is the propagation phase shift picked up when the wave is propagating in Dielectric 2. Then we get

$$\psi_{34} = \frac{4\pi}{\lambda} n_4 d_4. \tag{9}$$

Since $|r_{3,45}|$ increases with $n_4$ (see Supplementary Fig. 1), materials of high refractive index should be used as Dielectric 2 to achieve the maximum value of $|r_{3,45}|$ and its thickness can be calculated using Eq. (9). It is interesting to notice that $\psi_{34} = \arctan\left[ 2\kappa_3 n_4 / \left( \kappa_3^2 - n_4^2 \right) \right]$ decreases with wavelength due to the increase of $\kappa_3$ with the wavelength for Ag, and the value of $4\pi n_4 d_4/\lambda$ also decreases with the wavelength. Therefore, Eq. (9) can be potentially self-fulfilled over a broad spectrum, resulting in broadband high transmittance.

Finally, the refractive index and the thickness of Dielectric 1 are suitably selected for the optimal transmittance. The phasor diagram in Fig. 2b also gives

$$\psi_{23} - \psi_{3,45} = \pi, \tag{10}$$

$$\psi_{23} + \frac{4\pi}{\lambda} n_2 d_2 - \psi_{12} = \pi. \tag{11}$$

Substituting $\psi_{12} = \pi$ into Eq. (11) gives

$$\psi_{23} + \frac{4\pi}{\lambda} n_2 d_2 = 0. \tag{12}$$

$4\pi n_2 d_2/\lambda$ decreases with the wavelength and $\psi_{23} = \arctan\left[ 2\kappa_3 n_2 / \left( \kappa_3^2 - n_2^2 \right) \right]$ also decreases with the wavelength because $\kappa_3$ increases with the wavelength for Ag. Unlike Eq. (9), Eq. (12) cannot be satisfied over a broad spectrum. Thus, to achieve broadband high transmittance, we need to sweep the refractive index and the thickness of Dielectric 1 within a small range to find out the optimal value. Since the refractive index of typical dielectrics are between ~1.38 (magnesium fluoride, $MgF_2$) and ~2.6 (titanium dioxide, $TiO_2$), the required efforts are significantly reduced compared to the case where the

refractive indices and thicknesses of all the three D–M–D layers need to be optimized.

Now, we summarize the design procedure for highly transparent DMD electrodes. Step 1 (select $n_3$, $\kappa_3$, and $d_3$): choose a low-loss and highly conductive metallic film for the DMD layer (in most cases, use Ag). The metallic film thickness is designed to achieve the electrical conductivity required in practical applications. It is worth noting that higher overall transmittance can be achieved by using a thinner metallic film. Step 2 (select $n_4$): use high refractive index materials for Dielectric 2. The larger $n_4$ is used, the higher overall transmittance can be obtained. Step 3 (determine $d_4$): the thickness of Dielectric 2 is calculated using Eq. (9). Step 4 (design $n_2$ and $d_2$): optimize the refractive index and the thickness of Dielectric 1 within a small range to achieve the optimal broadband transmittance.

**Proof of concept.** First, we use the results for an ideal DMD structure at a single wavelength (Fig. 2c) reported in a previous study[39] to verify the design guidance provided in Eqs. (9), (10), and (12). According to ref. [39], an ideal DMD can be achieved at a selected wavelength for $n_1 = n_5 = 1$, $n_3 = 0$, $n_2 = n_4 = \kappa_3$, and $d_2 = d_4 = \lambda/(8\kappa_3)$. Substituting $n_4 = \kappa_3$ into Eq. (9) gives $d_4 = \lambda/(8\kappa_3)$, substituting $n_2 = \kappa_3$ into Eq. (12) gives $d_2 = \lambda/(8\kappa_3)$, and substituting $n_4 = \kappa_3$ into Eq. (10) gives $n_2 = n_4$, all of which are perfectly consistent with the conclusions obtained in ref. [39].

In realistic cases, the substrate cannot be air (i.e., $n_5 \neq n_1$) and the refractive indices of common dielectrics are usually much smaller than the extinction coefficients of metals in the visible range (i.e., $n_4 \ll \kappa_3$). This implies that different materials are required for Dielectrics 1 and 2 in a realistic DMD structure targeting optimal broadband transparency, i.e., $n_2 \neq n_4$. However, the same material was used for Dielectrics 1 and 2 in most of the previous studies (more than 70 published works[28]), which means those results are not optimal in terms of optical transmittance and indicates the necessity of providing quantitative guidelines in order to obtain optimized DMD electrode designs.

Next, the optical performance of a DMD electrode on a common optical substrate, PET, is designed based on our proposed guidelines. The flexible PET substrate exhibits an averaged absolute transmittance of ~88.0% from 400 to 700 nm as shown in Supplementary Fig. 2. A 9-nm-thick Ag film is employed as the middle metallic layer and a $TiO_2$ film with recorded-high $n_4 = 2.6$ reported in ref. [27] is used for Dielectric 2. At the center wavelength of the visible range (i.e., 550 nm), $\psi_{34}$ is calculated as 1.37, which leads to Dielectric 2 thickness $d_4 = 24$ nm according to Eq. (9). As displayed in Fig. 2d, Eq. (9) is satisfied over the entire visible spectrum at $d_4 = 24$ nm because both the left ($\psi_{34}$) and right ($4\pi n_4 d_4/\lambda$) terms of the equation decreases with the wavelength, which is consistent with our prediction in the above discussion. As summarized in Fig. 2e, a more straightforward way to find out the optimal $d_4$ value at which the DMD structure presents the highest averaged transmittance in the visible range (400–700 nm) is to sweep $d_4$ within a range (e.g., from 10 to 50 nm) and obtain the best transmittance at each $d_4$ value by sweeping the refractive index ($n_2$) and the thickness ($d_2$) of Dielectric 1. As seen from the plot, the highest averaged absolute transmittance of ~89.8% is achieved at $d_4 = 24$ nm, which agrees perfectly with the calculated results using Eq. (9) and thereby validates the accuracy of our proposed method. Correspondingly, the refractive index and the thickness of Dielectric 1 are $n_2 = 1.7$ and $d_2 = 63$ nm, respectively, when the highest transmittance is achieved at $d_4 = 24$ nm. The absolute transmittance and reflectance spectra of the optimized structure, i.e., PET substrate/24 nm $TiO_2$/9 nm Ag/63 nm Dielectric 1, spanning the entire visible range is provided in Supplementary

Fig. 3. This $n_2$ can be realized by $Al_2O_3$ film fabricated using common deposition methods. The relative transmittance of the optimized DMD electrode $T_r$ is ~102.0% compared to the simulated transmittance (~88.0%) of the PET substrate, which indicates that our proposed design procedure effectively guarantees the optimal optical performance of the DMD structure and a relative transmittance >100% is achievable with suitable material selection and structure design.

**Experimental demonstrations.** Based on the previous analysis, high transparency can be achieved by suppressing the metal absorption loss with thin Ag films. However, large surface roughness is an unavoidable issue for physical-vapor-deposited (evaporated or sputtered) ultra-thin (<20 nm) Ag films due to Ag atom's intrinsic three-dimensional (3D) Volmer–Weber growth mode[40]. As an example, the root-mean-square (RMS) roughness of an evaporated 15 nm Ag film can be as large as 6 nm[41]. Such an issue will affect the film conductivity and induce additional scattering loss, thereby impairing both electrical and optical performance of the DMD electrodes.

We recently demonstrated a novel approach to achieve ultra-thin (down to 6 nm) and ultra-smooth (RMS roughness < 1 nm) Ag film by doping a metallic element (e.g., Aluminum, Copper, Titanium, or Chromium) during Ag deposition (as illustrated in Supplementary Fig. 4)[2,5,29,42]. In this work, we choose Cu-doped Ag as the metallic layer as this film offers the lowest optical loss among different doped Ag films[5]. The deposition rates of Cu and Ag were chosen as 0.19 and 11.09 Å s$^{-1}$, respectively, which corresponds to ~2% Cu atomic concentration. The added Cu atoms effectively suppress the 3D island formation of the Ag atoms during film deposition and promote an early-stage formation of ultra-thin (<10 nm) Ag films. An atomic force microscopy (AFM) image showing details of a 6.5-nm-thick Cu-doped Ag film deposited on a fused silica substrate is displayed in Fig. 3a. The RMS roughness of the film is measured to be ~0.47 nm, which is more than 10 times lower than that of pure Ag at a similar thickness reported in previous works[2,5,41]. A 3D view of the AFM scan is further provided in Supplementary Fig. 5 to show the surface morphology and roughness of the ultrathin Cu-doped Ag film.

As a direct representation of the light absorption property of a film, the imaginary part ($\varepsilon_2$) of the measured permittivity ($\varepsilon = \varepsilon_1 + i\varepsilon_2 = (n + i\kappa)^2$) of the 6.5-nm-thick Cu-doped Ag (Fig. 3b) is very close to that of pure Ag in the visible range (400–700 nm), which indicates the low optical loss of the ultrathin Cu-doped Ag film. Here, a 20 nm-thick pure Ag film was sputter-deposited for

the calibration and the refractive indices of both Cu-doped Ag and pure Ag films were characterized using the spectroscopic ellipsometry. In addition, the sheet resistance of this thin Cu-doped Ag film was measured as ~18.6 Ω sq$^{-1}$, showing the high conductivity of the ultrathin Cu-doped Ag film.

Next, a DMD-based transparent electrode is constructed employing this 6.5-nm-thick Cu-doped Ag. Here, ~50-μm-thick PET film, which exhibits excellent mechanical flexibility, is used as the substrate. Zinc oxide (ZnO) is selected as Dielectric 2 due to its high refractive index ($n \sim 2.0$) and negligible optical loss in the visible range (ZnO is the highest refractive index material achievable in our group). The refractive indices of PET, ZnO, and Cu-doped Ag across the whole visible wavelength range are provided in Supplementary Fig. 6. The thickness ($d_4$) of Dielectric 2 is calculated as ~24 nm with Eq. (9) considering $n_4 \sim 2.05$ and $\tilde{n}_3 \sim 0.19 + 3.39i$ at 550 nm wavelength. As shown in Fig. 4a, the interesting compensation between the left ($\psi_{34}$) and right ($4\pi n_4 d_4/\lambda$) sides of Eq. (9) spanning 400–700 nm is also effectively verified in this case. Thus, 24 nm ZnO is an optimal selection not only for a single wavelength (550 nm) but also for the entire visible range. As the last step, the refractive index and the thickness of the top dielectric (Dielectric 1) is swept from 1.3 to 2.6 and from 1 to 100 nm with a step of 0.05 and 1 nm, respectively, to find out the optimal combination for the highest averaged absolute transmittance across the 400–700 nm wavelength range. Figure 4b shows that $n_2 = 1.65$ and $d_2 = 56$ nm for the optimal condition and this $n_2$ is close to the refractive index of $Al_2O_3$. By applying the measured refractive index of $Al_2O_3$ (Supplementary Fig. 6), $d_2$ was re-optimized and is still equal to 56 nm.

As illustrated in the inset in Fig. 4c, the final configuration of the DMD transparent electrode is determined as PET substrate/ 24 nm ZnO/6.5 nm Cu-doped Ag/56 nm $Al_2O_3$. The measured and simulated optical spectra of the designed electrode are presented in Fig. 4c, showing great consistency with each other. The averaged absolute transmittances across 400–700 nm calculated from experimental and theoretical results are ~(88.4 ± 0.1)% and ~88.4%, respectively. Figure 4c clearly shows that the transmittance of the DMD electrode is higher than the absolute transmittance of the bare PET substrate (~(88.1 ± 0.4)% from the measurement) with the optical transmittance from 416 to 607 nm effectively enhanced with the two antireflection dielectrics. Here, the cited uncertainties above represent one standard deviation of the measured data. The experimental averaged transmittances of the DMD electrode and the PET substrate are calculated with the measurement results of 15 samples. Detailed data can be found in Supplementary Figs. 2 and 7. In contrast to all the results reported in previous works, the relative transmittance of our proposed

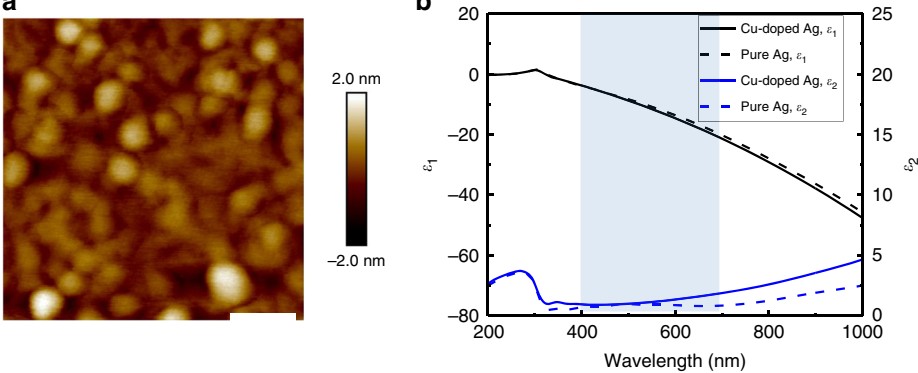

**Fig. 3 Characterizations of a Cu-doped Ag thin film. a** Atomic force microscopy (AFM) characterizations of a Cu-doped Ag thin film. The scale bar is 100 nm. **b** Measured relative permittivity ($\varepsilon = \varepsilon_1 + \varepsilon_2$) values as a function of free-space wavelength of Cu-doped Ag and pure Ag. The shaded region refers to the visible range (400–700 nm), over which our DMD electrode is optimized.

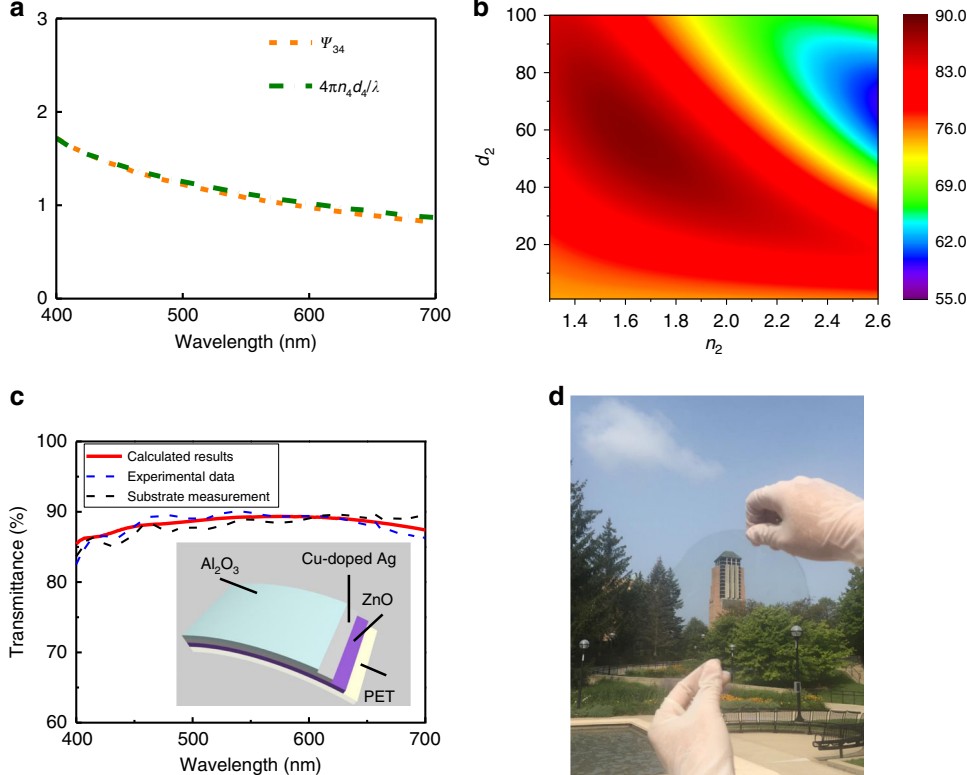

**Fig. 4 Experimental demonstration of the designed DMD electrode. a** The calculated values of the left and right terms in Eq. (9) when using 24 nm ZnO and 6.5 nm Cu-doped Ag as Dielectric 2 and the metallic layer, respectively, in the DMD structure. It shows Eq. (9) is well satisfied across the entire visible range. **b** Averaged absolute visible transmittance (%) of the DMD electrode dependent on the refractive index and the thickness of the top dielectric (Dielectric 1) when Dielectric 2 and the metallic layer are selected as 24 nm ZnO and 6.5 nm Cu-doped Ag, respectively. **c** Calculated (red solid curve) and measured (blue dashed curve) absolute transmittance from 400 to 700 nm of the designed DMD transparent electrode, showing great consistency with each other. The representative measured spectrum with an averaged transmittance of ~88.4% is taken from the measurement results of 15 samples. As a comparison, one representative transmittance spectrum of the bare PET substrate with the averaged transmittance of ~88.1% is provided to show how the optical performance of the optimized DMD electrode gets improved. Inset presents the configuration of the designed DMD electrode, i.e., PET substrate/ 24 nm ZnO/6.5 nm Cu-doped Ag/56 nm $Al_2O_3$. **d** A photograph of the fabricated flexible electrode, showing high transparent and neutral appearance.

DMD electrode surpasses 100% ($T_r$ ~ 100.3% based on the measurement data) as its optical loss (~11.6%) resulting from the stack reflection and absorption (=100%−transmittance) is well suppressed below the ~11.9% optical loss of the bare PET substrate (Supplementary Fig. 2). The angular performance of the DMD electrode is further investigated in Supplementary Fig. 8, which shows that high transmittance >75% is maintained up to 60° angle of incidence. A photograph of the fabricated flexible DMD structure on the PET substrate is displayed in Fig. 4d, which clearly shows the high transparency.

It is worthwhile noting that even higher transmittance can be achieved by using a bottom dielectric of higher refractive index than that of ZnO, such as $TiO_2$ (whose refractive index is about 2.6 in the visible range[27,43,44]). In this case, the thickness of the high-index $TiO_2$ layer is determined as $d_4 = 22$ nm using Eq. (9). The refractive index and the thickness of Dielectric 1 are optimized as $n_2 = 1.6$ and $d_2 = 72$ nm, respectively, by sweeping $n_2$ and $d_2$ within a broad range. The optimal structure 'PET substrate/22 nm $TiO_2$/6.5 nm Cu-doped Ag/72 nm Dielectric 1' gives out an averaged absolute transmittance of 89.6% across 400–700 nm, which corresponds to a relative transmittance of ~101.8% as presented in Supplementary Fig. 9.

Finally, we discuss the functionality of our designed DMD electrode and other material options depending on the intended applications. Firstly, the use of the insulating $Al_2O_3$ layer will make the proposed DMD structure an excellent candidate for the transparent heat mirror[28], that can be used for window defrosting

or deicing applications. Secondly, to use the DMD structure as an electrode in optoelectronic devices, the $Al_2O_3$ layer can be replaced by other dielectric materials featuring a similar refractive index that can also function as an effective electron/hole transport layer. Possible candidates include ZnO sol–gel films[1], which are commonly used as electron transport layers in solar cells and organic LEDs (OLEDs). Due to its porous structure, the refractive index is much lower than that of a dense ZnO film prepared by vacuum-deposition methods and is very close to the ideal refractive index ($n_2 = 1.65$) required in our design (Supplementary Fig. 10). Notably, our proposed design principles are not limited to treating the ambient as air, in Supplementary Information we show that they are also applicable when designing a DMD electrode used in a solar cell, as an example of optoelectronic devices (see Supplementary Fig. 11 for more details). Thirdly, the dielectric layers (ZnO and $Al_2O_3$) also play a critical role in protecting the ultrathin metallic film from degradation. As presented in Supplementary Fig. 12, the DMD electrode on the PET substrate has survived the accelerated test under high temperature and humidity (85 °C, 85% relative humidity), showing ~12.6% change of the sheet resistance after 120 h test, while the sheet resistance of the ultrathin Cu-doped Ag without any protections increases quickly to infinity under the same test condition after 8 h.

## Discussion

Relative transmittance higher than 100% was achieved in this work by the integration of a novel Cu-doped Ag film into an

optimized DMD structure. This Cu-doped Ag film features ultra-thin thickness (~6.5 nm thick), ultra-smooth morphology (roughness < 1 nm), low optical loss, and high electrical conductivity (sheet resistance ~18.6 Ω sq$^{-1}$), and was fabricated using a room-temperature deposition method. The optimized DMD structure was designed by quantitative principles that were generalized for the first time in this work. We experimentally demonstrated that the flexible DMD-based electrode, although not optimal, has 88.4% absolute averaged transmittance over the visible spectrum, which is higher than 88.1% transmittance of its PET substrate. This study provides an exciting pathway to address the major challenge faced by existing flexible transparent electrodes and to replace traditional ITO counterparts, thus facilitating high-performance flexible optoelectronic devices.

## Methods

**Film deposition**. Materials used in this work were all deposited by sputtering (LAB 18, Kurt J. Lesker Co.). Cu-doped Ag films were deposited by DC co-sputtering from Cu and Ag targets with 4.5 mTorr argon (Ar) pressure. The optimized deposition rates of Cu and Ag were 0.19 and 11.09 Å s$^{-1}$. Pure Ag was deposited by DC sputtering for 12.4 Å s$^{-1}$ and 4.5 mTorr condition. ZnO and Al$_2$O$_3$ films were deposited by RF sputtering with 4.5 and 3.0 mTorr Ar pressure, respectively.

**Material and optical characterizations**. All optical simulations were performed using the transfer matrix method. The refractive indices and thicknesses of materials and transmission spectra of electrodes were measured by spectroscopic ellipsometry method (M-2000, J. A. Woollam Inc.). The reflection spectra of the fabricated electrodes were measured by a thin-film measurement instrument (F20, Filmetrics) integrated with a spectrometer and light source. The sheet resistances were measured using a four-point probe method (FPP-5000, Miller Design & Equipment). The surface morphology of Cu-doped Ag films was characterized by tapping mode atomic force microscopy (Dimension Icon AFM, Bruker Corporation).

## Data availability

The data that supports the findings of this study are available from the corresponding author upon reasonable request. The source data underlying Supplementary Figs. 2 and 7 are provided as a Source Data file.

## Code availability

Transfer matrix method codes used for optical simulations are available from the corresponding author upon reasonable request.

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

## Acknowledgements

D.L. acknowledges the support by the Basic Science Center Program for Ordered Energy Conversion of the National Natural Science Foundation of China (Grant No. 51888103), NSFC (Grant No. 51976090), and Fundamental Research Funds of the Central Universities (No. 30919011403). C.J. and L.J.G. acknowledges the MTRAC Advanced Materials Hub 2018 Award. C.Z. would like to acknowledge the start-up funding from Huazhong University of Science and Technology. C.J. acknowledges helpful discussions with Mr. Kaito Yamada, Mr. Zhong Zhang, and Dr. Heyan Wang.

## Author contributions

C.J., D.L., and L.J.G. conceived the project. C.J. and D.L. are responsible for theoretical design and simulation. C.J. and C.Z. performed the experimental work. D.L. and L.J.G. directed the project. All authors discussed the results and contributed to the manuscript.

## Competing interests

C.Z. and L.J.G. declare financial interest in the ultra-thin metal-based transparent conductor technology.
