## [Peer Review File · Nature Communications]

Reviewers' comments:

Reviewer #1 (Remarks to the Author):

The paper presents a design method and an experimental proof-of-concept for transparent dielectric/metal/dielectric DMD electrodes with relative optical transmittance over 100%. The results presented in the paper are certainly interesting for the community. Nevertheless, the paper has weak points that cannot be overseen. These are:

1. The existing literature is not adequately cited. Characteristically, only for the metal meshes/nanowires the authors cite 10 papers (line 43 in the manuscript), which is more than the citations concerning the DMD electrodes. Many important papers are thus left uncited. A couple of important review papers on the DMD electrodes are:
 - Cao et al., Transparent electrodes for organic optoelectronic devices: a review, *J. of Photonics for Energy*, 4 (2014) 040990;
 - Zilberberg and Riedl, Metal-nanostructures - a modern and powerful platform to create transparent electrodes for thin-film photovoltaics, *J. Mater. Chem. A Mater. Energy Sustain.* 4, (2016) 14481
2. The authors claim that to the best of their knowledge, finding design rules for the quantification and optimization of the transmittance over a broad spectrum in the visible has not been addressed before (line 89-94). In fact, many papers exist that tackle this issue.

Apart from the optimization for single wavelength using admittance method, e.g. in the cited paper Kostlin and Frank, Optimization of transparent heat mirrors based on a thin silver film between antireflection films, *Thin Solid Films* 89 (1982) 287, or in Grosse et al., Design of low emissivity systems based on a three-layer coating, *J. Non-Cryst. Solids* 218 (1997) 38), there are different papers tackling electrode design for broadband transparency using transfer matrix methods. Some relevant papers here are:

- Kim and Lee, Design of dielectric/metal/dielectric transparent electrodes for flexible electronics, *J. of Photonics for Energy* 2 (2012) 021215;
- Ham et al., Design of broadband transparent electrodes for flexible organic solar cells, *J. Mater. Chem. A* 1 (2013) 3076;
- Bauch and Dimopoulos, Design of ultrathin metal-based transparent electrodes including the impact of interface roughness, *Mater. Des.* 104 (2016) 37.

The design rules appearing in the present paper (regarding the choice of the metal, the bottom and top dielectric refractive index, thickness, etc) agree with the design rules that have been drawn in the literature. So, there is a question about the novelty of the present design approach.

3. The effect of a higher DMD transmittance for the electrode-plus-substrate compared to the plain substrate has been reported in the literature, at least in: Kinner et al, *Materials and Design* 168 (2019) 107663, for a broad window in the visible.
4. The realized electrode of ZnO/Cu:Ag/Al₂O₃, despite its very high transparency, cannot be easily applied to devices like solar cells, OLED etc. The question is how would the carriers from the active device layers be efficiently extracted through the thick insulating Al₂O₃ layer. Also, the authors do not elaborate on how the device architecture would alter the electrode design, which is a critical issue in such electrodes.
5. The authors do not discuss to the needed extent the influence of the layer roughness (metal and

dielectric) on the optical and electrical properties of the electrode. They have included SEM images of the metal on a fused silica substrate that is much smoother than the PET substrate they finally use for experimental demonstration. Also, the metal is deposited on ZnO finally, which has its own roughness and influences the metal percolation threshold. Large roughness will affect the optical properties and should be considered in electrode design.

In conclusion, although I think the results are interesting and worth reporting, I do not consider they consist a considerable advancement in the state of the art and, due to the mentioned weaknesses, I regret to advise that the paper is not suitable for publication in Nature Communications. After addressing the points above, the results certainly worth publishing in another journal.

Reviewer #2 (Remarks to the Author):

Report on the manuscript: "Ultrathin-metal-film-based transparent electrodes with relative transmission surpassing 100%".

Today, due to the growing demand for transparent electrodes and flexible optoelectronic components, it is necessary to look for new solutions to make these electrodes. Indeed, ITO is the dominant transparent electrode nowadays but indium is rare and ITO is not flexible. It is therefore necessary to look for "Indium free transparent electrodes". The introduction of the submitted article constitutes a particularly well-written state of the art concerning this field of investigation.

-Just a comment about the bibliography. A great deal of works have already been published in this field, and also reviews summarizing these publications, some of which should be mentioned, such as: C. Guillén and J. Herrero, TCO/metal/TCO structures for energy and flexible electronics, Thin Solid Films 520 (2011) 1-17.

Part 2.1 and 2.2 -The rigour of the authors' scientific approach must be stressed: first, a well-reasoned and developed theoretical treatment, followed by a transition to an experimental level which makes it possible to corroborate the theoretical study. Figures 1 and 2 are well documented and commented, so that the theoretical development is well understood. However, some facts have already been known for a long time, such as the need to use high refractive index dielectrics... Nevertheless, the method used in this work makes it possible to grasp the coherence of the choices made, which is rarely the case in the earlier works published in this field.

Part 2.3 -After estimating through theoretical study the optimum structure DMD', the theoretical results are faced with the results of the experimental study. Nevertheless, between the theoretical ideal and the harsh reality of things there are often important differences. Thus, it is well known that the growth of Ag thin films follows the Volmer-Weber growth mode, which means that Ag films are neither homogeneous nor smooth, so that they exhibit a high surface roughness. As well, it is necessary to deposit a relatively large thickness of silver in order to obtain the coalescence, and hence the conductivity, of the silver clusters resulting from the three-dimensional growth of the silver layer. Such relatively large thickness of silver is unfavourable to good transmission of visible light This bottleneck has been over passed by authors of the present work. Indeed, they have previously shown that the introduction of a small percentage of Cu into the Ag layer (~2% Cu atomic concentration) allows two-dimensional growth, which permits achieving coalescence for very thin metal layer. Finally, it is shown that structures such as "PET substrate/24 nm ZnO/6.5nm Cu-doped Ag/56 nm Al₂O₃" allows achieving relative transmittance which surpasses 100% (Tr ~ 100.5%). Such result is quite spectacular and deserves real interest for its originality. Nevertheless, keeping in mind that this structure is being built to replace ITO transparent electrodes, one must question its functionality. Indeed, if the longitudinal sheet resistance of this electrode is 18.6 Ω/sq, one can

question its transversal conductivity. The thickness of the top alumina layer, which is a perfect insulator, is 56 nm and the probability for an electron to pass through such an insulating layer thickness by tunnel effect is very low. Therefore using this electrode as bottom layer in optoelectronic device will probably result in a very poor performing device. In contrast, such structures should be very efficient as transparent heat mirror. Do these electrodes have been tested in optoelectronic components?

-Have the authors probed the stability of their structures? Recent publications have shown that the stability of the DMD structures in the presence of Cu depends strongly on the nature of the dielectric used, if Cu does not diffuse in ZnO what is it with Al₂O₃?

In conclusion, once the authors have responded to these comments, the manuscript will deserve publication.

J C Bernède

Reviewer #3 (Remarks to the Author):

Summary

This manuscript pertains to the development of new transparent conducting electrodes (TCEs), which have importance in optoelectronic devices. It advocates for using the dielectric-metal-dielectric (DMD) structure for next-generation TCEs, and proposes a quantitative design principle to expedite structure engineering of these DMD TCEs; in particular, it points out that there is no reason why the two dielectrics should be the same, and develops equations governing optimal material and film thickness choices. This development follows the principles of anti-reflective (AR) coatings (though without using that language) accounting for complex refractive indices. It then demonstrates the process experimentally using an ultra-thin Cu-doped Ag metal electrode that has been previously developed by the group. Using their principles, they improve film transmittance from ~88.1% to ~88.4% over the visible spectrum (400-700 nm).

Major Comments

The proposed design principle simplifies the thin film engineering for DMD TCEs by providing constraints on the search space, particularly in light of them pointing out that the two dielectrics need not be the same material. The majority of work in DMD TCEs uses the same dielectric on both sides of the metal, so highlighting this is valuable to the community; it would be helpful to place this insight in a more prominent place in the paper. While the principles of their theoretical development are all accessible at an undergraduate level, they have nicely laid out the constraints and reduced the effort to engineer a film based on the DMD structure.

The quantitative portion of the paper would be clearer if the authors used the language of AR coatings; the proposed DMD structure for a TCE seems built around this idea, and the enhanced transmission of an optical film with an AR coating--even with a reflective metal interface--is well established. While it may not be trivial to optimize, any optical engineer is familiar with the principle to improve transmittance. The opening of section 2.1 could be much more concise by referencing AR principles then moving to the particular structure for the design constraints. If there is a clear argument as to why their structure is not an AR coating (or an application of the principle) it should be explicit. If the structure is, in fact, an application of AR coating principles, the title and article should be revised to reflect this.

In the experimental portion, while the low RMS roughness of the ultra-thin Cu-doped Ag film is indeed impressive, the authors should make clear that this is not the first report of this film. The authors have previously reported similar results on a film in ACS Appl. Mater. Interfaces 2019, 11, 27216–27225, which was not mentioned in the context of the low RMS roughness, instead making the comparison to a pure Ag film (line 248). The exposition in section 2.3 could be considerably simplified and shortened by referencing this prior work.

The claim of all previous work on metallic-material-based TCEs showing relative transmittance of <100% is dubious (e.g. lines 19-21). A quick search reveals, for instance, NPG Asia Materials, 10, 309–317 (2018), which reports improved transmittance (using the AR language) with a sheet resistance of <20 ohm / sq. The authors should revise their claims to account for this and other developments.

For many optoelectronic devices, the transmittance vs. angle is important. Calculations and experimental data for transmittance vs. angle would be very useful to evaluate the potential performance of this structure in device applications.

Detailed Notes

Figure 3a: Scale of grains in SEM and AFM are different. Explain. What new information does the SEM provide that the AFM does not?

Figure 3b: Enhance contrast to show the structure more clearly.

Figure 3c: What is the shaded region? I assume this is meant to show the visible range over which the device is being optimized, but it is not clear.

Figure 4c: What are error bars on experimental data? Zoom in on transmittance axis to show fluctuations in experiment and theory

Figure S3: It would be helpful to see the modeled transmittance of bare PET in this figure as well.

Figure S4: This schematic has been published multiple times (e.g. ACS Appl. Mater. Interfaces, 11, 27216–27225 (2019)). Explicit attribution may be necessary.

Figure S5: As the thin, low RMS film deposition is already published, is this figure necessary?

As this electrode is ultimately for use in a device, the quantitative derivation would be more useful if n_1 is allowed to be arbitrary (i.e. the index can be set to the device rather than air). This would permit the DMD electrode to be adjusted to a particular application.

It would be useful to have Figure 4c and Figure S2 on the same plot to evaluate how the optimized structure performs at different wavelengths relative to the bare PET substrate. For instance, at 400nm, the structure performs worse than the PET, and overall there appears to be more curvature. While this might be expected, the noted broadband transmittance (lines 150-154) should be evident with this data.

Line 34: replace "by" with "because of" for the sentence meaning

Lines 279-280: Can the "consistency" be quantified? Do the modeled and experimental transmittance agree within error?

Reviewer #1:

The paper presents a design method and an experimental proof-of-concept for transparent dielectric/metal/dielectric DMD electrodes with relative optical transmittance over 100%.

The results presented in the paper are certainly interesting for the community. Nevertheless, the paper has weak points that cannot be overseen. These are:

1. The existing literature is not adequately cited. Characteristically, only for the metal meshes/nanowires the authors cite 10 papers (line 43 in the manuscript), which is more than the citations concerning the DMD electrodes. Many important papers are thus left uncited. A couple of important review papers on the DMD electrodes are:

- Cao et al., Transparent electrodes for organic optoelectronic devices: a review, *J. of Photonics for Energy*, 4 (2014) 040990;
- Zilberberg and Riedl, Metal-nanostructures - a modern and powerful platform to create transparent electrodes for thin-film photovoltaics, *J. Mater. Chem. A Mater. Energy Sustain.* 4, (2016) 14481

→ We thank the reviewer for pointing out these references that we overlooked. We have added many more related references in the revised text:

31. Cao, W., Li, J., Chen, H. & Xue, J. Transparent electrodes for organic optoelectronic devices: a review. *J. Photonics Energy* 4, 040990 (2014).

32. Zilberberg, K. & Riedl, T. Metal-nanostructures – a modern and powerful platform to create transparent electrodes for thin-film photovoltaics. *J. Mater. Chem. A* 4, 14481-14508 (2016).

33. Kim, S. & Lee, J.-L. Design of dielectric/metal/dielectric transparent electrodes for flexible electronics. *J. Photonics Energy* 2, 021215 (2012).

34. Ham, J., Kim, S., Jung, G. H., Dong, W. J. & Lee, J.-L. Design of broadband transparent electrodes for flexible organic solar cells. *J. Mater. Chem. A* 1, 3076-3082 (2013).

35. Bauch, M. & Dimopoulos, T. Design of ultrathin metal-based transparent electrodes including the impact of interface roughness. *Mater. Des.* 104, 37-42 (2016).

36. Kinner, L. et al. Polymer interlayers on flexible PET substrates enabling ultra-high performance, ITO-free dielectric/metal/dielectric transparent electrode. *Mater. Des.* 168, 107663 (2019).

37. Guillén, C. & Herrero, J. TCO/metal/TCO structures for energy and flexible electronics. *Thin Solid Films* 520, 1-17 (2011).

2. The authors claim that to the best of their knowledge, finding design rules for the quantification and optimization of the transmittance over a broad spectrum in the visible has not been addressed before (line 89-94). In fact, many papers exist that tackle this issue.

Apart from the optimization for single wavelength using admittance method, e.g. in the cited paper Kostlin and Frank, Optimization of transparent heat mirrors based on a thin silver film between antireflection films, *Thin Solid Films* 89 (1982) 287, or in Grosse et al., Design of low emissivity systems based on a three-layer coating, *J. Non-Cryst. Solids* 218 (1997) 38), there are different papers tackling electrode design for broadband transparency using transfer matrix methods. Some relevant papers here are:

- Kim and Lee, Design of dielectric/metal/dielectric transparent electrodes for flexible electronics, *J. of Photonics for Energy* 2 (2012) 021215;
- Ham et al., Design of broadband transparent electrodes for flexible organic solar cells, *J. Mater. Chem. A* 1 (2013) 3076;
- Bauch and Dimopoulos, Design of ultrathin metal-based transparent electrodes including the impact of interface roughness, *Mater. Des.* 104 (2016) 37.

The design rules appearing in the present paper (regarding the choice of the metal, the bottom and top dielectric refractive index, thickness, etc) agree with the design rules that have been drawn in the literature. So, there is a question about the novelty of the present design approach.

→ We thank the reviewer for pointing out these references. After going over them, we respectfully express disagreement with the reviewer’s question on the novelty of our work. To emphasize the novelty and the importance of our presented design approach, we comment on above three works and show that the designs in them either rely on exhaustive searching method, *i.e.*, not a “quantitative” design rule, or are questionable and thereby not as effective as what we laid out in the manuscript.

(1) In Section 2.2 of the paper by Kim and Lee, the WO₃/Ag/WO₃ DMD structure was designed. First, the same material, WO₃, was used for both dielectric layers, which cannot achieve optimal performance according to our analysis and optimization results. Secondly, the thicknesses of one WO₃ layer and the Ag layer were exhaustively optimized with the thickness of the other WO₃ layer fixed at 30 nm. Therefore, fundamental quantitative design guidelines were not provided in the paper by Kim and Lee.

(2) In the paper by Ham et al., the Dielectric 1/Ag/Dielectric 2/substrate DMD structure was designed as follows. First, the refractive index and thickness of Dielectric 1 were exhaustively optimized for the Dielectric 1/Ag/substrate structure with the thickness of Ag fixed at 10 nm. Then, using the optimized refractive index and thickness of Dielectric 1, the refractive index and thickness of Dielectric 2 were optimized to maximize the transmittance of the Dielectric 1/Ag/Dielectric 2/substrate structure.

Below, we repeat their process to obtain the highest transmittance, T_1 , and then use our design principles to obtain T_2 . We will show $T_2 > T_1$, which proves that their method is not effective and optimal as our design principles.

The substrate is glass ($n_s=1.5$) and the thickness of Ag is 10 nm with its complex refractive index taken from Ref. [38]. At each refractive index of Dielectric 1, n_{D1} , from 1.5 to 3.0, its thickness, d_{D1} , was swept to find out the optimal Dielectric 1/Ag/glass structure that has the highest transmittance averaged over the spectrum from 400 nm to 700 nm. Figure R1a shows that the optimal condition is $n_{D1} = 2.3$ and $d_{D1} = 24$ nm. Then, using these optimal parameters, the optimal refractive index, n_{D2} , and thickness, d_{D2} , of Dielectric 2 were optimized to be $n_{D2} = 1.7$ and $d_{D2} = 61$ nm as shown in Figure R1b. The highest averaged transmittance, T_1 , is ~ 88.5%.

Following our design principles, TiO₂ with recorded-high $n_{D2} = 2.6$ reported in Ref. [27] was used for Dielectric 2. $d_{D2} = 24$ nm obtained from Equation (9). Then n_{D1} and d_{D1} were swept and the optimal condition is $n_{D1} = 1.7$ and $d_{D1} = 63$ nm. The highest averaged transmittance, T_2 , is 91.6%, which is considerably higher than T_1 . Therefore, the design in the paper by Ham et al. is questionable, but clearly not as effective as ours.

Figure R1. Optimization of the Dielectric 1/Ag/ Dielectric 2/glass DMD structure following the process reported in the paper by Ham et al. (a) Variation of the averaged transmittance of the Dielectric 1/Ag/glass structure with n_{D1} . At each n_{D1} from 1.5 to 3.0, d_{D1} was swept to find out the optimal Dielectric 1/Ag/glass structure that has the highest transmittance averaged over the spectrum from 400 nm to 700 nm. (b) Variation of the averaged transmittance of the Dielectric 1/Ag/ Dielectric 2/glass structure with n_{D2} for optimal $n_{D1} = 2.3$ and $d_{D1} = 24$ nm. (c) Transmittance spectra of the

structures designed following the process reported in the paper by Ham *et al.* and following our proposed principles.

(3) In the paper by Bauch and Dimopoulos, the AZO/Au/TiO_x/glass DMD structure was designed. First, the metal, Au, is not the best choice due to its higher optical loss than Ag. Secondly, AZO and TiO_x were selected without explaining the reasons. Thirdly, the thicknesses of AZO and TiO_x were exhaustively optimized. Therefore, quantitative design guidelines were not provided in the paper by Bauch and Dimopoulos.

List of changes

1) In the revised text, we modified the description in Introduction as following: “This counter-intuitive, yet achievable performance is obtained by (i) quantitative design principles that are generalized in this work, particularly, analytic expression of the optimal bottom dielectric thickness and analytical result showing that different materials should be used for the two dielectrics...”

2) We added all the above three works into the references and compared them with our results in the revised text:

33. Kim, S. & Lee, J.-L. Design of dielectric/metal/dielectric transparent electrodes for flexible electronics. *J. Photonics Energy* **2**, 021215 (2012).

34. Ham, J., Kim, S., Jung, G. H., Dong, W. J. & Lee, J.-L. Design of broadband transparent electrodes for flexible organic solar cells. *J. Mater. Chem. A* **1**, 3076-3082 (2013).

35. Bauch, M. & Dimopoulos, T. Design of ultrathin metal-based transparent electrodes including the impact of interface roughness. *Mater. Des.* **104**, 37-42 (2016).

3. The effect of a higher DMD transmittance for the electrode-plus-substrate compared to the plain substrate has been reported in the literature, at least in: Kinner et al, *Materials and Design* **168** (2019) 107663, for a broad window in the visible.

→ We thank the reviewer for mentioning this work. We have carefully read the above paper and confirmed that the reported relative transmittance averaged over the spectrum from 400 nm to 700 nm is less than 100%, even though TiO_x, a dielectric with higher refractive index than that in our paper, was used in the above paper. The reported averaged transmittance of DMD on PET substrate is 85.1% over the spectrum from 400 nm to 700 nm, while the averaged transmittance of the bare PET substrate is higher than 86%. The transmittance of DMD is higher than PET from 483 nm to 622 nm, but much lower in the rest of the spectrum. See Figure 5a of the above paper for more details.

In revisions, we added the above work into the references:

36. Kinner, L. et al. Polymer interlayers on flexible PET substrates enabling ultra-high performance, ITO-free dielectric/metal/dielectric transparent electrode. *Mater. Des.* **168**, 107663 (2019).

4. The realized electrode of ZnO/Cu:Ag/Al₂O₃, despite its very high transparency, cannot be easily applied to devices like solar cells, OLED etc. The question is how would the carriers from the active device layers be efficiently extracted through the thick insulating Al₂O₃ layer. Also, the authors do not elaborate on how the device architecture would alter the electrode design, which is a critical issue in such electrodes.

→ We thank the reviewer for the valuable comments. How to apply our DMD in real optoelectronic devices is also a question we considered when coming up with the designs.

(1) First, we would like to comment on the material selections when incorporating our DMD electrode into an optoelectronic device. In our paper, Al₂O₃ was selected as the dielectric layer to prove that our DMD electrode can achieve a transmittance higher than that of the substrate itself just because its refractive index is very close to the ideal value ($n_2 = 1.65$) obtained using our theory.

When incorporating our proposed structure into a real optoelectronic device (e.g., solar cells or OLEDs), the Al_2O_3 layer can be replaced by other materials featuring a similar refractive index that can also function as an effective electron/hole transport layer. Possible candidates include ZnO sol-gel films, which are commonly used as electron transport layers in solar cells and OLEDs. Due to the porous structure, its refractive index is much lower than that of a dense ZnO film prepared by vacuum-deposition methods and is very close to the ideal refractive index ($n_2 = 1.65$) required in our DMD electrode. The following figure compares the refractive indices of ZnO sol-gel film used in Ref. [S2] to that of the Al_2O_3 film used in this work. It is clear that their refractive indices are very close to each other. When replacing the top Al_2O_3 layer in our proposed DMD electrode with the ZnO sol-gel film, the averaged transmittance from 400 to 700 nm is calculated as $\sim 88.3\%$, which is very close to that of the Al_2O_3 -based electrode ($\sim 88.4\%$) and is higher than that of the PET substrate ($\sim 88.0\%$).

Figure S10. (a) The refractive index of the ZnO sol-gel film extracted from Ref. [S2] compared to that of the Al_2O_3 film in this work. Both of them are very close to the ideal refractive index ($n_2 = 1.65$) required in our design. (b) Calculated transmittance of the DMD electrode after replacing the top Al_2O_3 with the ZnO sol-gel film. The averaged transmittance from 400 to 700 nm is $\sim 88.3\%$, which is very close to that of the Al_2O_3 -based electrode ($\sim 88.4\%$) and is higher than that of the PET substrate ($\sim 88.0\%$).

(2) Next, we show here that that our presented design principles are still applicable when designing a DMD electrode used in a semi-transparent solar cell, an example of optoelectronic devices. The simplified, yet representative solar cell structure is shown in Figure S11. A ternary blend heterojunction was used as the active layer. The thicknesses of ITO, the active layer, and Ag are 145 nm, 85 nm and 16 nm, respectively. The complex refractive indices of ITO and the active layer were taken from Ref. [S2] and the complex refractive index of Ag was taken from Ref. [38]. See Ref. [S2] for more details on the solar cell structure.

Since the active layer selectively absorbs sunlight in the red and near-infrared spectra, the transmittance of the semi-transparent solar cell averaged over the spectrum from 400 nm to 700 nm, T_{Device} , can still be used as the figure of merit for the DMD design. We will show that the highest transmittance obtained by following our design principles agrees well with that obtained by exhaustively optimizing the refractive indices and thicknesses of Dielectrics 1 and 2.

The highest averaged transmittance was $\sim 55.4\%$ obtained from exhaustive sweep of the refractive indices and thicknesses of Dielectrics 1 and 2 with the transmittance spectrum shown in Figure S11. Following our design principles, TiO_2 with recorded-high $n_6 = 2.6$ reported in Ref. [27] was used for Dielectric 2. $d_6 = 24$ nm obtained from Equation (9). Then n_{D1} and d_{D1} was swept to find out the optimal parameters to achieve the highest averaged transmittance. The optimal condition, where $n_4 = 2.1$ and $d_4 = 51$ nm, gives the highest averaged transmittance of $\sim 54.9\%$, which agrees well with that

designed by the exhaustive method. Figure S11 also shows the transmittance spectra of the respective structures designed by different methods and the two spectra are nearly identical.

Regarding designing the DMD electrode that replaces ITO in an opaque solar cell, the DMD electrode is usually independently designed to maximize its transmittance and then directly integrated into the solar cell in many literatures including the one (*J. Mater. Chem. A* **4**, 14481, 2016) suggested by the reviewer.

The results demonstrate the applicability of our design principles for designing a DMD electrode used in an optoelectronic device.

Figure S11. The structure and the transmittance spectra of a semi-transparent solar cell. The refractive indices and thicknesses of Dielectrics 1 and 2 were designed by both the principles described in this work and the exhaustive method.

List of changes

1) We added the above figures as Figure S10 and Figure S11.

2) We added the following statements in the main text: “Finally, we discuss the functionality of our designed DMD electrode and other material options depending on the intended applications. Firstly, the use of the insulating Al_2O_3 layer will make the proposed DMD structure an excellent candidate for the transparent heat mirror, that can be used for window defrosting or deicing applications. Secondly, to use the DMD structure as an electrode in optoelectronic devices, the Al_2O_3 layer can be replaced by other dielectric materials featuring a similar refractive index that can also function as an effective electron/hole transport layer. Possible candidates include ZnO sol-gel films, which are commonly used as electron transport layers in solar cells and organic LEDs (OLEDs). Due to its porous structure, the refractive index is much lower than that of a dense ZnO film prepared by vacuum-deposition methods and is very close to the ideal refractive index ($n_2 = 1.65$) required in our design (Figure S10). Notably, our proposed design principles are not limited to treating the ambient as air, in Supplementary Information we show that they are also applicable when designing a DMD electrode used in a solar cell, as an example of optoelectronic devices (see Figure S11 for more details).”

5. The authors do not discuss to the needed extent the influence of the layer roughness (metal and dielectric) on the optical and electrical properties of the electrode. They have included SEM images of the metal on a fused silica substrate that is much smoother than the PET substrate they finally use for experimental demonstration. Also, the metal is deposited on ZnO finally, which has its own

roughness and influences the metal percolation threshold. Large roughness will affect the optical properties and should be considered in electrode design.

→ We thank the reviewer for highlighting this point. As shown in the figure below, we characterized the surface morphologies of a bare PET substrate and a PET film coated with a 24 nm ZnO layer using AFM. The RMS roughness values of the PET substrate and the ZnO-coated PET are measured to be ~ 0.72 nm and ~ 1.10 nm, respectively. Although the ZnO-coated PET presents a slightly higher roughness than the silica substrate, the optical property of the ultrathin Cu-doped Ag film deposited on ZnO/PET is not affected too much, which can be verified by the high consistency between the measured and simulated (simulation is performed using the measured refractive index of a thin Cu-doped Ag film deposited on a silica substrate) transmittance of the final DMD structure on the PET substrate in Figure 4c.

Figure R2. Atomic force microscopy (AFM) characterizations of (a) a bare PET substrate and (b) a PET film coated with 24 nm ZnO. The scale bars are 100 nm. The RMS roughness values of the PET substrate and the ZnO-coated PET are measured to be ~ 0.72 nm and ~ 1.10 nm, respectively.

We also agree with the reviewer’s comment that the rough surface of either the PET substrate or the ZnO-coated PET would impact the properties of the ultrathin Cu-doped Ag to some extent. However, it is not an easy task to accurately characterize the optical properties of the ultrathin Cu-doped Ag film on a flexible and thin ZnO-coated PET film using the spectroscopic ellipsometry. We plan to conduct more careful and detailed studies regarding this topic in a separate work in the near future.

In conclusion, although I think the results are interesting and worth reporting, I do not consider they consist a considerable advancement in the state of the art and, due to the mentioned weaknesses, I regret to advice that the paper is not suitable for publication in Nature Communications. After addressing the points above, the results certainly worth publishing in another journal.

→ We have revised the manuscript to further clarify the novelty of our proposed design approach and of our demonstration of relative transmittance surpassing 100%. The novelty of our work is also supported by Reviewers #2 and #3.

Reviewer #2:

Report on the manuscript: “Ultrathin-metal-film-based transparent electrodes with relative transmission surpassing 100%”.

Today, due to the growing demand for transparent electrodes and flexible optoelectronic components, it is necessary to look for new solutions to make these electrodes. Indeed, ITO is the dominant transparent electrode nowadays but indium is rare and ITO is not flexible. It is therefore necessary to look for “Indium free transparent electrodes”. The introduction of the submitted article constitutes a particularly well-written state of the art concerning this field of investigation.

→ We thank the reviewer for the positive comment.

-Just a comment about the bibliography. A great deal of works have already been published in this field, and also reviews summarizing these publications, some of which should be mentioned, such as: C. Guillén and J. Herrero, TCO/metal/TCO structures for energy and flexible electronics, *Thin Solid Films* 520 (2011) 1-17.

→ We thank the reviewer for pointing out the oversight. We have added the following references in the revised text:

37. Guillén, C. & Herrero, J. TCO/metal/TCO structures for energy and flexible electronics. *Thin Solid Films* 520, 1-17 (2011).

Part 2.1 and 2.2 -The rigour of the authors' scientific approach must be stressed: first, a well-reasoned and developed theoretical treatment, followed by a transition to an experimental level which makes it possible to corroborate the theoretical study. Figures 1 and 2 are well documented and commented, so that the theoretical development is well understood. However, some facts have already been known for a long time, such as the need to use high refractive index dielectrics... Nevertheless, the method used in this work makes it possible to grasp the coherence of the choices made, which is rarely the case in the earlier works published in this field.

Part 2.3 –After estimating through theoretical study the optimum structure DMD', the theoretical results are faced with the results of the experimental study. Nevertheless, between the theoretical ideal and the harsh reality of things there are often important differences. Thus, it is well known that the growth of Ag thin films follows the Volmer–Weber growth mode, which means that Ag films are neither homogeneous nor smooth, so that they exhibit a high surface roughness. As well, it is necessary to deposit a relatively large thickness of silver in order to obtain the coalescence, and hence the conductivity, of the silver clusters resulting from the three-dimensional growth of the silver layer. Such relatively large thickness of silver is unfavourable to good transmission of visible light This bottleneck has been over passed by authors of the present work. Indeed, they have previously shown that the introduction of a small percentage of Cu into the Ag layer (~2% Cu atomic concentration) allows two-dimensional growth, which permits achieving coalescence for very thin metal layer.

Finally, it is shown that structures such as “PET substrate/24 nm ZnO/6.5nm Cu-doped Ag/56 nm Al₂O₃” allows achieving relative transmittance which surpasses 100% (Tr ~ 100.5%).

→ We thank the reviewer for the recognition of our work's value and the positive comments on the novelty of our work.

Such result is quite spectacular and deserves real interest for its originality. Nevertheless, keeping in mind that this structure is being built to replace ITO transparent electrodes, one must question its functionality. Indeed, if the longitudinal sheet resistance of this electrode is 18.6 Ω /sq, one can question its transversal conductivity. The thickness of the top alumina layer, which is a perfect insulator, is 56 nm and the probability for an electron to pass through such an insulating layer thickness by tunnel effect is very low. Therefore using this electrode as bottom layer in optoelectronic device will probably result in a very poor performing device. In contrast, such structures should be very efficient as transparent heat mirror. Do these electrodes have been tested in optoelectronic components?

→ We thank the reviewer for the very valuable comment. How to apply our DMD in real optoelectronic devices is a question we considered when coming up with the designs. In fact, Al₂O₃ was selected as the dielectric layer to prove that our DMD electrode can achieve a transmittance higher than that of the substrate itself just because its refractive index is very close to the ideal value ($n_2 = 1.65$) obtained using our theory. When incorporating our proposed structure into a real optoelectronic device (e.g., solar cells or OLEDs), the Al₂O₃ layer can be replaced by other materials featuring a similar refractive index that can also function as an effective electron/hole transport layer. Possible candidates include ZnO sol-gel films, which are commonly used as electron transport layers in solar cells and OLEDs. Due to the porous structure, its refractive index is much lower than that of a dense

ZnO film prepared by vacuum-deposition methods and is very close to the ideal refractive index ($n_2 = 1.65$) required in our DMD electrode. The following figure compares the refractive indices of ZnO sol-gel film used in Ref. [S2] to that of the Al_2O_3 film used in this work. It is clear that their refractive indices are very close to each other. When replacing the top Al_2O_3 layer in our proposed DMD electrode with the ZnO sol-gel film, the averaged transmittance from 400 to 700 nm is calculated as $\sim 88.3\%$, which is very close to that of the Al_2O_3 -based electrode ($\sim 88.4\%$) and is higher than that of the PET substrate ($\sim 88.0\%$). In addition, our presented ultrathin Ag alloy films have been successfully used in optoelectronic devices including OLEDs (*ACS Appl. Mater. Interfaces* **11**, 27216, 2019) and transparent solar cells (*Adv. Mater.* **31**, 1903173, 2019). Therefore, our designed DMD electrodes should work well in optoelectronic devices.

Figure S10. (a) The refractive index of the ZnO sol-gel film extracted from Ref. [S2] compared to that of the Al_2O_3 film in this work. Both of them are very close to the ideal refractive index ($n_2 = 1.65$) required in our design. (b) Calculated transmittance of the DMD electrode after replacing the top Al_2O_3 with the ZnO sol-gel film. The averaged transmittance from 400 to 700 nm is $\sim 88.3\%$, which is very close to that of the Al_2O_3 -based electrode ($\sim 88.4\%$) and is higher than that of the PET substrate ($\sim 88.0\%$).

In revisions, we added the above figure as Figure S10 and the following statements in the main text: “Finally, we discuss the functionality of our designed DMD electrode and other material options depending on the intended applications. Firstly, the use of the insulating Al_2O_3 layer will make the proposed DMD structure an excellent candidate for the transparent heat mirror, that can be used for window defrosting or deicing applications. Secondly, to use the DMD structure as an electrode in optoelectronic devices, the Al_2O_3 layer can be replaced by other dielectric materials featuring a similar refractive index that can also function as an effective electron/hole transport layer. Possible candidates include ZnO sol-gel films, which are commonly used as electron transport layers in solar cells and organic LEDs (OLEDs). Due to its porous structure, the refractive index is much lower than that of a dense ZnO film prepared by vacuum-deposition methods and is very close to the ideal refractive index ($n_2 = 1.65$) required in our design (Figure S10).”

-Have the authors probed the stability of their structures? Recent publications have shown that the stability of the DMD structures in the presence of Cu depends strongly on the nature of the dielectric used, if Cu does not diffuse in ZnO what is it with Al_2O_3 ?

→ We thank the reviewer for bringing up this critical point. As mentioned by the reviewer, the stability of Cu (and also Ag) depends largely on the outside dielectrics in the DMD structure. We did the accelerated stability test of our DMD electrode on the PET substrate under high temperature and humidity (85°C and 85% relative humidity). As depicted in the figure below, the sheet resistance of

the DMD electrode shows a change of only ~ 12.6% after 120 hours test, which indicates that the Cu-doped Ag film is well protected by ZnO and Al₂O₃ layers. In contrast, an ultrathin Cu-Ag without any dielectric protections degrades quickly and its sheet resistance becomes infinity after 8 hours in the same test.

Figure S12. Accelerated humidity test results of DMD-based transparent electrodes as a function of test duration. The test condition is 85°C and 85% relative humidity.

In revisions, we added the above figure as Figure S12 and the following statements in the main text: “Thirdly, the dielectric layers (ZnO and Al₂O₃) also play a critical role in protecting the ultrathin metallic film from degradation. As presented in Figure S12, the DMD electrode on the PET substrate has survived the accelerated test under high temperature and humidity (85°C, 85% relative humidity), showing ~ 12.6% change of the sheet resistance after 120 hours test, while the sheet resistance of the ultrathin Cu-Ag without any protections increases quickly to infinity under the same test condition after 8 hours.”

In conclusion, once the authors have responded to these comments, the manuscript will deserve publication.

→ We have responded to all the reviewer’s comments and revised the manuscript correspondingly.

Reviewer #3:

Summary

This manuscript pertains to the development of new transparent conducting electrodes (TCEs), which have importance in optoelectronic devices. It advocates for using the dielectric-metal-dielectric (DMD) structure for next-generation TCEs, and proposes a quantitative design principle to expedite structure engineering of these DMD TCEs; in particular, it points out that there is no reason why the two dielectrics should be the same, and develops equations governing optimal material and film thickness choices. This development follows the principles of anti-reflective (AR) coatings (though without using that language) accounting for complex refractive indices. It then demonstrates the process experimentally using an ultrathin Cu-doped Ag metal electrode that has been previously developed by the group. Using their principles, they improve film transmittance from ~88.1% to ~88.4% over the visible spectrum (400-700 nm).

Major Comments

The proposed design principle simplifies the thin film engineering for DMD TCEs by providing constraints on the search space, particularly in light of them pointing out that the two dielectrics need not

be the same material. The majority of work in DMD TCEs uses the same dielectric on both sides of the metal, so highlighting this is valuable to the community; it would be helpful to place this insight in a more prominent place in the paper. While the principles of their theoretical development are all accessible at an undergraduate level, they have nicely laid out the constraints and reduced the effort to engineer a film based on the DMD structure.

→ We thank the reviewer for the positive comment. In the revised text, we modified the description in Introduction as following: “This counter-intuitive, yet achievable performance is obtained by (i) quantitative design principles that are generalized in this work, particularly, analytic expression of the optimal bottom dielectric thickness and analytical result showing that different materials should be used for the two dielectrics...”

The quantitative portion of the paper would be clearer if the authors used the language of AR coatings; the proposed DMD structure for a TCE seems built around this idea, and the enhanced transmission of an optical film with an AR coating--even with a reflective metal interface--is well established. While it may not be trivial to optimize, any optical engineer is familiar with the principle to improve transmittance. The opening of section 2.1 could be much more concise by referencing AR principles then moving to the particular structure for the design constraints. If there is a clear argument as to why their structure is not an AR coating (or an application of the principle) it should be explicit. If the structure is, in fact, an application of AR coating principles, the title and article should be revised to reflect this.

→ We thank the reviewer for this valuable comment. In revisions, we modified the description in Introduction as following: “In recent years, dielectric-metal-dielectric (DMD) based transparent electrodes have been noted as potential alternatives. In this type of electrodes, a thin metallic film is **sandwiched between two antireflection dielectrics** to induce high transparency. They feature high transparency and conductivity, low haze, excellent flexibility, facile fabrication, and great compatibility with different substrates.”

We modified the description in Section 2.1 as following: “However, in this section, we will provide guidelines built **around the antireflection principles** to reduce R_1 and A using the DMD structure, and show that with optimized design ($R_1 + A$) can be made smaller than the single side reflection of the substrate, R_2 , leading to relative transmittance surpassing 100%.”

Also, we added the following statements in Section 2.3: “Figure 4c clearly shows that the transmittance of the DMD electrode is higher than the absolute transmittance of the bare PET substrate ($\sim (88.1 \pm 0.4) \%$ from the measurement) with the optical transmittance from 416 to 607 nm effectively enhanced with the **two antireflection dielectrics**.”

In the experimental portion, while the low RMS roughness of the ultra-thin Cu-doped Ag film is indeed impressive, the authors should make clear that this is not the first report of this film. The authors have previously reported similar results on a film in *ACS Appl. Mater. Interfaces* 2019, 11, 27216–27225, which was not mentioned in the context of the low RMS roughness, instead making the comparison to a pure Ag film (line 248). The exposition in section 2.3 could be considerably simplified and shortened by referencing this prior work.

→ We thank the reviewer for highlighting this point. In revisions, we removed the SEM images showing the surface morphologies of the Cu-doped Ag film (the original Figure 3a) and the pure Ag (the original Figure S5), and cited the related references when describing the surface roughness issue of the pure Ag film instead. We also simplified the corresponding description: “An atomic force microscopy (AFM) image showing details of a 6.5-nm-thick Cu-doped Ag film deposited on a fused silica substrate is displayed in Figure 3a. The RMS roughness of the film is measured to be ~ 0.47 nm, which is more than 10 times lower than that of pure Ag at a similar thickness reported in previous works.^{2, 5, 41} A 3D view of the AFM scan is further provided in Figure S5 to show the surface morphology and roughness of the ultrathin Cu-doped Ag film.”

(Note: Reference 2 is *Adv. Mater.* **26**, 5696, 2014, Reference 5 is *ACS Appl. Mater. Interfaces* **11**, 27216, 2019, and Reference 41 is *Nano Lett.* **9**, 178 2009.)

The claim of all previous work on metallic-material-based TCEs showing relative transmittance of <100% is dubious (e.g. lines 19-21). A quick search reveals, for instance, NPG Asia Materials, 10, 309–317 (2018), which reports improved transmittance (using the AR language) with a sheet resistance of <20 ohm / sq. The authors should revise their claims to account for this and other developments.

→ We went over the cited reference, and respectfully have to disagree with the reviewer’s comment. In the paper mentioned by the reviewer (*NPG Asia Mater.* **10**, 309–317, 2018), they pre-deposited anti-reflection (AR) coatings that consist of alternative high- and low-index layers on both sides of the PET substrate, forming the so-called ‘ARPET’, to enhance the transparency of the substrate itself. The details can be found in Table 1S in the SI of the paper. Then, the final transparent electrode was achieved by patterning CuO/Cu meshes on one side of the ARPET (please refer to Figure S5 in SI). Although the transmittance of the CuO/Cu meshes on ARPET is higher than that of the bare PET, the relative transmittance should be calculated by comparing the transmittances of the final electrode and the ARPET, which is < 100%. In addition, the transmittance of the CuO/Cu meshes on bare PET is also lower than that of the bare PET, rendering relative transmittance < 100%. In fact, it is really challenging to achieve transparent electrodes with the relative transmittance > 100%, no matter what kind of schemes are used. The relative transmittance of transparent electrodes based on metallic networks is unavoidably limited to < 100% due to the scattering and blocking effects of the metallic nanowires or meshes. The relative transmittances of DMD-based transparent electrodes reported previously are also < 100%, which is limited by the availability of ultrathin metallic film of low optical loss and non-optimal selections of the two dielectric layers. However, there are tons of studies on transparent electrodes and we may have missed some works that happened to achieve a relative transmittance > 100% when reviewing the area of transparent electrodes as reminded by the reviewer. Therefore we changed this claim in the abstract: “Thus, to obtain practical sheet resistance values, the visible transmittance of the electrodes in previous studies is typically lower than the transparent substrates the electrode structures are built on, *i.e.*, the transmittance relative to the substrate is < 100%.”

For many optoelectronic devices, the transmittance vs. angle is important. Calculations and experimental data for transmittance vs. angle would be very useful to evaluate the potential performance of this structure in device applications.

→ We thank the reviewer for the valuable suggestion. We re-evaluated the optical performance at large incident angles and found out that > 75% transmittance can be well maintained up to 60° angle of incidence as showing in the figure below.

Figure S8. Measured and calculated transmittance spectra of the DMD electrode on the PET substrate at different incident angles. High transmittance > 75% is maintained up to 60° angle of incidence. The slight discrepancy between the simulations and measurements at short wavelengths is due to the small index fitting inaccuracy of the PET substrate, which is discussed in Figure S2.

In revisions, we added the above figure as Figure S8 and the following statements in the main text: “The angular performance of the DMD electrode is further investigated in Figure S8, which shows that high transmittance > 75% is maintained up to 60° angle of incidence.”

Detailed Notes

Figure 3a: Scale of grains in SEM and AFM are different. Explain. What new information does the SEM provide that the AFM does not?

→ We thank the reviewer for the suggestion. As mentioned by the reviewer, since the AFM image can provide the information of both surface morphology and roughness, we removed the SEM image and the corresponding description in the revised text.

Figure 3b: Enhance contrast to show the structure more clearly.

→ We thank the reviewer for the suggestion. We have reduced the range of the scale bar to enhance the figure contrast and updated the following figure in Figure 3 in the revised main text.

Figure 3. (a) Atomic force microscopy (AFM) characterizations of a Cu-doped Ag thin film. The scale bar is 100 nm. (b) Measured relative permittivity ($\epsilon = \epsilon_1 + i\epsilon_2$) values as a function of free-space wavelength of Cu-doped Ag and pure Ag. The shaded region refers to the visible range (400 – 700 nm), over which our DMD electrode is optimized.

In addition, we have added the following 3D view of the AFM scan as Figure S5 in the revised SI to show the surface morphology and roughness of the 6.5-nm-thick Cu-doped Ag film.

Figure S5. 3D view of the atomic force microscopy (AFM) scan of the 6.5-nm-thick Cu-doped Ag film deposited on a fused silica substrate. The root-mean-square (RMS) roughness of the film is measured to be ~ 0.47 nm.

Figure 3c: What is the shaded region? I assume this is meant to show the visible range over which the device is being optimized, but it is not clear.

→ We thank the reviewer for pointing out the oversight. In revisions, we added the following statements in the captions of Figure 3: “The shaded region refers to the visible range (400 – 700 nm), over which our DMD electrode is optimized.”

Figure 4c: What are error bars on experimental data? Zoom in on transmittance axis to show fluctuations in experiment and theory.

→ We thank the reviewer for pointing it out. We re-collected the transmittance spectra of 15 DMD electrodes and 15 PET substrate samples, and calculated the standard deviation of the measurement results.

In revisions, we included the new measurement results in Figure S2 and S7 and added the following statements in the main text: “The averaged absolute transmittances across 400 – 700 nm calculated from experimental and theoretical results are $\sim (88.4 \pm 0.1) \%$ and $\sim 88.4\%$, respectively. Figure 4c clearly shows that the transmittance of the DMD electrode is higher than the absolute transmittance of the bare PET substrate ($\sim (88.1 \pm 0.4) \%$ from the measurement) with the optical transmittance from 416 to 607 nm effectively enhanced with the two antireflection dielectrics. Here, the cited uncertainties above represent one standard deviation of the measured data. The experimental averaged transmittances of the DMD electrode and the PET substrate are calculated with the measurement results of 15 samples. Detailed data can be found in Figure S2 and S7.” In addition, we zoomed in the transmittance axis in Figure 4c to show the fluctuations of measured and calculated results as suggested by the reviewer.

Figure S3: It would be helpful to see the modeled transmittance of bare PET in this figure as well.

→ We thank the reviewer for the suggestion. In revisions, we added the simulated transmittance of the PET substrate into Figure S3 and updated the figure in SI.

Figure S3. Calculated absolute transmittance and reflectance spectra of the optimized structure ‘PET substrate / 24 nm TiO₂ / 9 nm Ag / 63 nm Dielectric 1’ across the entire visible range. Here, the refractive indices of TiO₂ and Dielectric 1 are assumed as 2.6 and 1.7, respectively. The averaged transmittance and reflectance are of ~ 89.8% and ~ 6.4%, respectively. The simulated transmittance spectrum of the PET substrate is provided as a reference, showing an averaged transmittance from 400 to 700 nm of ~ 88.0%. It is clear that the transmittance of the DMD electrode is well above that of the substrate especially at wavelengths < 600 nm.

Figure S4: This schematic has been published multiple times (e.g. ACS Appl. Mater. Interfaces, 11, 27216–27225 (2019)). Explicit attribution may be necessary.

→ We thank the reviewer for the suggestion. In revisions, we added the following statements and references in the captions of Figure S4: “More details of the deposition system can be found in our earlier work^{S1}.”

S1. Zhang, C. et al. An Ultrathin, Smooth, and Low-Loss Al-Doped Ag Film and Its Application as a Transparent Electrode in Organic Photovoltaics. *Adv. Mater.* **26**, 5696-5701 (2014).

Figure S5: As the thin, low RMS film deposition is already published, is this figure necessary?

→ We thank the reviewer for the valuable suggestion. In revisions, we removed Figure S5 from SI and the corresponding description from the main text. We also cited the related referenced in the main text when describing the surface roughness issue of the thin pure Ag film.

As this electrode is ultimately for use in a device, the quantitative derivation would be more useful if n_1 is allowed to be arbitrary (i.e. the index can be set to the device rather than air). This would permit the DMD electrode to be adjusted to a particular application.

→ We thank the reviewer for this valuable comment. We will show here that that our presented design principles are still applicable when designing a DMD electrode used in a semi-transparent solar cell, an example of optoelectronic devices. The simplified, yet representative solar cell structure is shown in Figure S11. A ternary blend heterojunction was used as the active layer. The thicknesses of ITO, the active layer, and Ag are 145 nm, 85 nm and 16 nm, respectively. The complex refractive indices of ITO and the active layer were taken from Ref. [S2] and the complex refractive index of Ag was taken from Ref. [38]. See Ref. [S2] for more details on the solar cell structure.

Since the active layer selectively absorbs sunlight in the red and near-infrared spectra, the transmittance of the semi-transparent solar cell averaged over the spectrum from 400 nm to 700 nm, T_{Device} , can still be used as the figure of merit for the DMD design. We will show that the highest

transmittance obtained by following our design principles agrees well with that obtained by exhaustively optimizing the refractive indices and thicknesses of Dielectrics 1 and 2.

The highest averaged transmittance was $\sim 55.4\%$ obtained from exhaustive sweep of the refractive indices and thicknesses of Dielectrics 1 and 2 with the transmittance spectrum shown in Figure S11. Following our design principles, TiO_2 with recorded-high $n_6 = 2.6$ reported in Ref. [27] was used for Dielectric 2. $d_6 = 24$ nm obtained from Equation (9). Then n_{D1} and d_{D1} was swept to find out the optimal parameters to achieve the highest averaged transmittance. The optimal condition, where $n_4 = 2.1$ and $d_4 = 51$ nm, gives the highest averaged transmittance of $\sim 54.9\%$, which agrees well with that designed by the exhaustive method. Figure S11 also shows the transmittance spectra of the respective structures designed by different methods and the two spectra are nearly identical.

Figure S11. The structure and the transmittance spectra of a semi-transparent solar cell. The refractive indices and thicknesses of Dielectrics 1 and 2 were designed by both the principles described in this work and the exhaustive method.

List of changes

1) We added the above figure as Figure S11.

2) We added the following statements in the main text: “Notably, our proposed design principles are not limited to treating the ambient as air, in Supplementary Information we show that they are also applicable when designing a DMD electrode used in a solar cell, as an example of optoelectronic devices (see Figure S11 for more details).”

It would be useful to have Figure 4c and Figure S2 on the same plot to evaluate how the optimized structure performs at different wavelengths relative to the bare PET substrate. For instance, at 400nm, the structure performs worse than the PET, and overall there appears to be more curvature. While this might be expected, the noted broadband transmittance (lines 150-154) should be evident with this data.

→ We thank the reviewer for the valuable suggestion. We added the measured transmittance spectra of the PET substrate into Figure 4c, showing how the optimized DMD electrode performs at different wavelengths across the visible range compared to the bare PET substrate.

In revisions, we updated the caption of Figure 4c and added the following statements in the main text: “Figure 4c clearly shows that the transmittance of the DMD electrode is higher than the absolute transmittance of the bare PET substrate ($\sim (88.1 \pm 0.4) \%$ from the measurement) with the optical transmittance from 416 to 607 nm effectively enhanced with the two antireflection dielectrics.”

Line 34: replace "by" with "because of" for the sentence meaning

→ We thank the reviewer for the suggestion. We have it corrected and highlighted in the revised manuscript.

Lines 279-280: Can the "consistency" be quantified? Do the modeled and experimental transmittance agree within error?

→ We thank the reviewer for pointing this out. As clarified in the above replies, we re-collected the transmittance spectra of 15 DMD electrodes and 15 PET substrate samples, and calculated the standard deviation of the measurement results. The averaged absolute transmittances across 400 – 700 nm calculated from experimental and theoretical results are $\sim (88.4 \pm 0.1) \%$ and $\sim 88.4\%$, respectively. It shows the difference between the measured and calculated values is within one standard deviation, based on which the 'consistency' can be quantified.

REVIEWERS' COMMENTS:

Reviewer #1 (Remarks to the Author):

The authors have responded satisfactorily in the majority of the comments raised in the reviews. The modifications made in the manuscript have increased its clarity. In view of these changes and the responses I can suggest the acceptance of the manuscript for publication.

Reviewer #2 (Remarks to the Author):

Th authors have responded to all the comments and the manuscript is now in order for publication.

Reviewer #3 (Remarks to the Author):

With these revisions, the message of the paper and its contributions are much clearer. In particular, the changes to Figures 4c and S3 are very useful to capture the improvement in transmission provided by the DMD architecture. I also appreciate the inclusion of the wider angle data (Figure S9) and the solar cell architecture (Figure S10) to evaluate the result for device applications.

One small change to fix a plural error on line 54: "in this type of electrodes" should be either "in this type of electrode" or "in these types of electrodes"

Reviewer #1:

The authors have responded satisfactorily in the majority of the comments raised in the reviews. The modifications made in the manuscript have increased its clarity. In view of these changes and the responses I can suggest the acceptance of the manuscript for publication.

→ We thank the reviewer for recommending publication of our manuscript.

Reviewer #2:

The authors have responded to all the comments and the manuscript is now in order for publication.

→ We thank the reviewer for recommending publication of our manuscript.

Reviewer #3:

With these revisions, the message of the paper and its contributions are much clearer. In particular, the changes to Figures 4c and S3 are very useful to capture the improvement in transmission provided by the DMD architecture. I also appreciate the inclusion of the wider angle data (Figure S9) and the solar cell architecture (Figure S10) to evaluate the result for device applications.

One small change to fix a plural error on line 54: "in this type of electrodes" should be either "in this type of electrode" or "in these types of electrodes"

→ We thank the reviewer for the positive comment. The text has been revised as “in this type of electrode”.